# The increasing threat to stratospheric ozone from dichloromethane

Ryan Hossaini[1], Martyn P. Chipperfield[2,3], Stephen A. Montzka[4], Amber A. Leeson[1], Sandip S. Dhomse[2,3] & John A. Pyle[5,6]

It is well established that anthropogenic chlorine-containing chemicals contribute to ozone layer depletion. The successful implementation of the Montreal Protocol has led to reductions in the atmospheric concentration of many ozone-depleting gases, such as chlorofluorocarbons. As a consequence, stratospheric chlorine levels are declining and ozone is projected to return to levels observed pre-1980 later this century. However, recent observations show the atmospheric concentration of dichloromethane—an ozone-depleting gas not controlled by the Montreal Protocol—is increasing rapidly. Using atmospheric model simulations, we show that although currently modest, the impact of dichloromethane on ozone has increased markedly in recent years and if these increases continue into the future, the return of Antarctic ozone to pre-1980 levels could be substantially delayed. Sustained growth in dichloromethane would therefore offset some of the gains achieved by the Montreal Protocol, further delaying recovery of Earth's ozone layer.

[1] Lancaster Environment Centre, Lancaster University, Lancaster LA1 4YQ, UK. [2] School of Earth and Environment, University of Leeds, Leeds LS2 9JT, UK. [3] National Centre for Earth Observation, University of Leeds, Leeds LS2 9JT, UK. [4] National Oceanic and Atmospheric Administration, Boulder, Colorado 80305, USA. [5] Department of Chemistry, University of Cambridge, Cambridge CB2 1EW, UK. [6] National Centre for Atmospheric Science, University of Cambridge, Cambridge CB2 1EW, UK. Correspondence and requests for materials should be addressed to R.H. (email: r.hossaini@lancaster.ac.uk).

I n the 1970s, it was recognized that chlorine and bromine released from long-lived anthropogenic compounds, such as chlorofluorocarbons (CFCs) and halons, could destroy ozone in the stratosphere[1,2]. Industrial emissions of these halocarbons have led to widespread depletion of Earth's ozone layer in recent decades, including the Antarctic 'Ozone Hole' phenomenon[3–5]. Peak ozone depletion was observed around the turn of the century, when globally the stratospheric ozone column was reduced by ∼5% relative to 1980 levels (a benchmark before which substantial ozone depletion had not been observed). While ozone depletion today remains a persistent environmental issue, there are signs that ozone recovery is underway[6–9], owing to controls on the production of ozone-depleting compounds introduced by the 1987 Montreal Protocol and its amendments[10,11]. Given compliance with the Protocol, stratospheric levels of chlorine derived from controlled ozone-depleting compounds should continue to steadily decline in coming decades. As a result, stratospheric column ozone is projected to return to pre-1980 levels in the middle to latter half of this century, depending on location[7,12].

Several human-produced chlorocarbons not controlled by the Montreal Protocol are present in Earth's atmosphere. Among the most abundant of these compounds is dichloromethane ($CH_2Cl_2$)—an industrial solvent also used as a feedstock in the production of other chemicals, among other applications[13,14]. Unlike CFCs, which are virtually inert in the troposphere and have long atmospheric lifetimes (decades to centuries), $CH_2Cl_2$ is a so-called very short-lived substance (VSLS)[15]. Historically, VSLS have been thought to play a minor role in stratospheric ozone depletion due to their relatively short atmospheric lifetimes (typically <6 months) and therefore low atmospheric concentrations. However, substantial levels of both natural and anthropogenic VSLS have been detected in the lower stratosphere[15–18] and numerical model simulations suggest a significant contribution of VSLS to ozone loss in this region[19–21]. Long-term measurements of $CH_2Cl_2$ reveal that its tropospheric abundance has increased rapidly in recent years[15,21–23]. For example, between 2000 and 2012, surface concentrations of $CH_2Cl_2$ increased at a global mean rate of almost 8% per year, with the largest growth observed in the Northern Hemisphere (NH)[21]. Given that natural emissions of $CH_2Cl_2$ are small, this recent growth likely reflects an increase in industrial emissions[15]. While the precise nature of the source remains poorly characterized, industrial $CH_2Cl_2$ emissions from Asia—in particular from the Indian subcontinent—appear to be growing in importance[23]. The impact of these observed changes and continued $CH_2Cl_2$ growth in coming decades on the timescale of stratospheric ozone recovery have not yet been considered.

Here, a state-of-the-art global chemical transport model (CTM) is used to examine the sensitivity of future stratospheric chlorine and ozone levels to sustained $CH_2Cl_2$ growth. CTMs are commonly used to investigate detailed chemical processes and here, because transport is specified in the model, we are able to isolate the ozone response solely to increasing levels of $CH_2Cl_2$ in the atmosphere. We find that continued growth in the atmospheric loading of $CH_2Cl_2$ could offset some of the future benefits of the Montreal Protocol and lead to a substantial delay (more than a decade) in the recovery of stratospheric ozone over Antarctica.

## Results

**Recent $CH_2Cl_2$ trends and future growth scenarios.** Figure 1 shows the measured surface abundance of $CH_2Cl_2$ from the National Oceanic and Atmospheric Administration (NOAA) long-term surface monitoring network. Globally, $CH_2Cl_2$ concentrations approximately doubled between 2004 and 2014

(Fig. 1a), although growth rates varied considerably during this period (Fig. 1b). The abundance of $CH_2Cl_2$ at mid-latitudes in the NH is around a factor of 3 greater than in the Southern Hemisphere (SH), reflecting NH industrial sources. At present, it is unknown if a single industrial application of $CH_2Cl_2$, or several, is contributing to the observed upward trend. As a common solvent, $CH_2Cl_2$ has numerous applications, which include use in metal cleaning/degreasing, in paint remover, and use by the pharmaceutical industry for preparing drugs. It is also used as blowing agent in production of foam plastics. A specific use of $CH_2Cl_2$, which seems likely to have increased in recent years, is in the manufacture of hydrofluorocarbons—the non-ozone-depleting chemicals used as replacements for CFCs and hydrochlorofluorocarbons (HCFCs). Given these sources, it is probable that demand for $CH_2Cl_2$ from developing countries now, and in coming years, will be relatively high. This is supported by elevated levels of $CH_2Cl_2$ detected over Asia, where Indian emissions are estimated to have increased by two- to fourfold between 1998 and 2008 (ref. 23).

Based on the NOAA surface measurements presented in Fig. 1a, we estimate a global emission source of around 1 Tg $CH_2Cl_2$ per year to sustain the observed $CH_2Cl_2$ concentrations in recent years (Fig. 2). We note that this is a far larger source than that of individual CFCs and other long-lived ozone-depleting gases (for example, carbon tetrachloride) in the 1980s, when emissions of those gases peaked. For $CH_2Cl_2$, and other VSLS more generally, relatively large emissions do not have the same impact on atmospheric concentrations, compared to say CFCs, as $CH_2Cl_2$ is more rapidly oxidized in the troposphere and has a much shorter atmospheric lifetime.

Two future scenarios encompassing potential surface $CH_2Cl_2$ increases from 2015 to 2100 have been derived and are considered in our forward model simulations. Both are based on observed long-term surface trends (Fig. 1c), and are designed to test the sensitivity of ozone to potential future changes in chlorine derived from $CH_2Cl_2$ growth. Scenario 1 assumes that surface $CH_2Cl_2$ continues to increase at the mean rate observed during the 2004–2014 period: 2.85 parts per trillion (p.p.t.) per year at mid-latitudes in the NH. Scenario 2, a more extreme growth scenario to test the sensitivity of ozone to larger $CH_2Cl_2$ increases, assumes $CH_2Cl_2$ continues to increase at the mean rate observed in the 2012–2014 period only: 6.1 p.p.t. per year. This period saw comparatively large $CH_2Cl_2$ growth compared to other recent years (Fig. 1b). In addition to the two growth scenarios, we also consider a third scenario in which no further $CH_2Cl_2$ growth occurs post 2016 (Methods section). In this scenario (Scenario 3), surface $CH_2Cl_2$ concentrations are fixed at 2016 levels throughout the forward simulation.

Constrained by the growth scenarios, our model simulations show a monotonic increase in chlorine from $CH_2Cl_2$ entering the stratosphere (Fig. 1d) in coming decades, from ∼70 p.p.t. Cl in 2014, to ∼180 p.p.t. Cl or ∼360 p.p.t. Cl by 2050, under Scenarios 1 or 2, respectively. Critically, the model reproduces well observed levels of $CH_2Cl_2$ around the tropopause in the recent past (Fig. 1d, inset) and, therefore, the stratospheric chlorine perturbation in response to increasing surface $CH_2Cl_2$ concentrations is realistic in our simulations.

**Impact of $CH_2Cl_2$ growth on stratospheric inorganic chlorine.** The dissociation of ozone-depleting compounds in the stratosphere liberates chlorine radicals which catalyse ozone loss. Owing to its relatively short stratospheric partial lifetime (of the order of 1–2 years in our model outside of the poles), $CH_2Cl_2$ dissociates rapidly and thereby makes its largest relative contribution to the pool of inorganic chlorine ($Cl_y$) in the lowermost stratosphere, at low latitudes (Fig. 3a–c). At present,

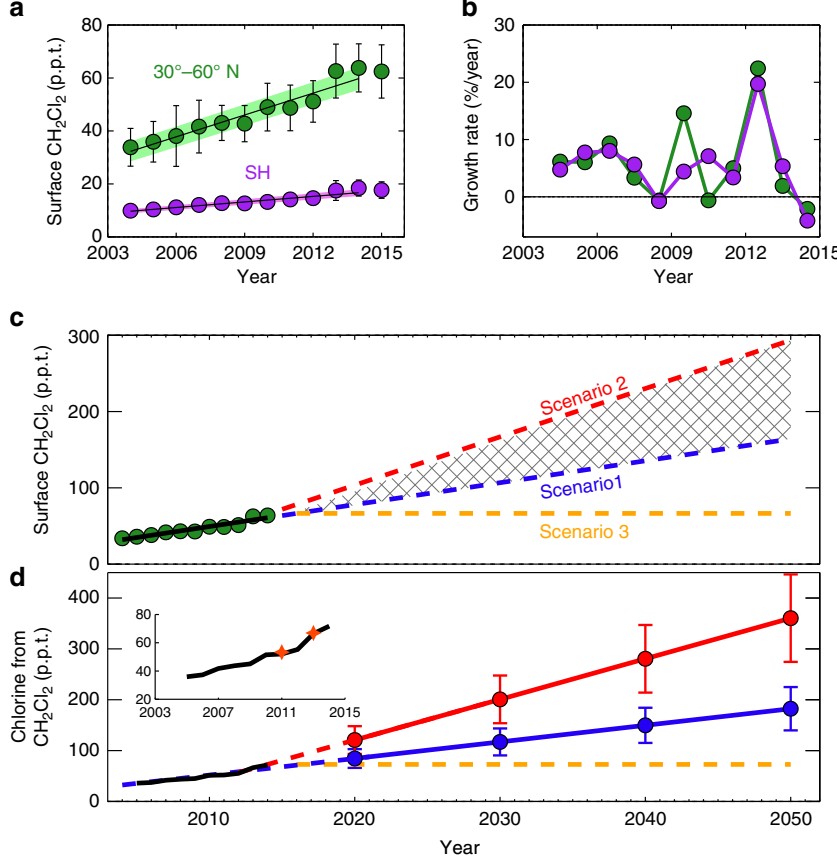

**Figure 1 | Observed trends and growth rate of surface CH₂Cl₂ and simulated stratospheric loading of chlorine.** (**a**) $CH_2Cl_2$ surface mixing ratio in p.p.t. from 2004 to 2015 derived from NOAA measurements as the annual mean observed at 4 sites in the SH, and 5 sites in the NH between 30° and 60° N (ref. 36). The time series is an update of ref. 21, years 2014 and 2015 are new data. Error bars denote ±1 s.d. and the solid lines denote a linear fit to these data with the shaded regions representing ±1 s.d. uncertainty on the fit. (**b**) Corresponding $CH_2Cl_2$ growth rates (% per year). (**c**) Observed surface $CH_2Cl_2$ mixing ratio in the NH (green circles, as in **a**) and trend (black line), along with projections of surface $CH_2Cl_2$ between 30 and 60° N latitude under future scenarios (dashed lines); $CH_2Cl_2$ increases at the mean rate observed over the 2004–2014 period (Scenario 1, blue), $CH_2Cl_2$ increases at the mean rate observed over the 2012–2014 period (Scenario 2, red) and $CH_2Cl_2$ remains at 2016 levels (Scenario 3, no future growth, orange). (**d**) Modelled chlorine (p.p.t.) from $CH_2Cl_2$ entering the stratosphere in the recent past and projections. This is derived by multiplying the simulated $CH_2Cl_2$ mixing ratio at the tropical tropopause by 2, to account for the 2 Cl atoms in the molecule. Data between 2005 and 2013 are an update of ref. 22, while subsequent years and future projections are from this study. Annual means in decadal intervals (2020–2050) are shown (filled circles) with ±1 s.d. (error bars) for Scenarios 1 (blue) and 2 (red). Solid lines denote a linear fit to these data, dashed portions extrapolate this fit prior to 2020. The orange line (dashed throughout) represents Scenario 3 (no future growth). Inset; Enlarged model curve for 2004–2014 with observed estimates from NASA aircraft measurements (stars).

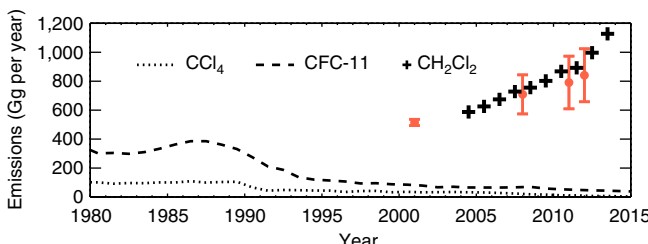

**Figure 2 | Time trend in global halocarbon emissions.** Emissions derived from a simple 1-box model for $CCl_4$ (dotted line), CFC-11 (dashed line) and $CH_2Cl_2$ (crosses) in units of Gigagrams (Gg) of source gas per year. Calculation for $CH_2Cl_2$ based on a parameterized global mean lifetime of 0.43 years. Also shown are recent independent estimates of $CH_2Cl_2$ emissions (orange points) from the AGAGE 12-box model[15]. Error bars denote uncertainty range.

$CH_2Cl_2$ accounts for <10% of stratospheric $Cl_y$, although this contribution would increase significantly in coming decades if $CH_2Cl_2$ growth continues and as chlorine from long-lived gases

decreases. By 2050 under Scenario 1, $CH_2Cl_2$ is projected to account for 20–30% of total $Cl_y$ in the lower stratosphere. An examination of the stratospheric $Cl_y$ trend in recent decades in our model reveals a peak in $Cl_y$ around the turn of the century at mid-latitudes (Fig. 3d,e). In the absence of $CH_2Cl_2$, here lower stratospheric $Cl_y$ is projected to return to pre-1980 levels by 2049, in line with ongoing decreases in levels of CFCs and other controlled long-lived $Cl_y$ precursors[7]. When $CH_2Cl_2$ growth is considered, this $Cl_y$ return date is delayed by around 15–17 years under Scenario 1. Under Scenario 2—an extreme scenario—the $Cl_y$ return date occurs after 2080.

The delay in the $Cl_y$ return date discussed above under Scenario 1 is significant and is additional to, and of a similar magnitude to, other factors which have previously been considered when assessing the uncertainty in return dates, for example, coupled chemistry-transport differences between climate models or different future greenhouse gas scenarios. The effect on the $Cl_y$ return date is also much larger than the influence of potentially eliminating remaining small levels of production or emission of CFCs and HCFCs[24]. In the upper

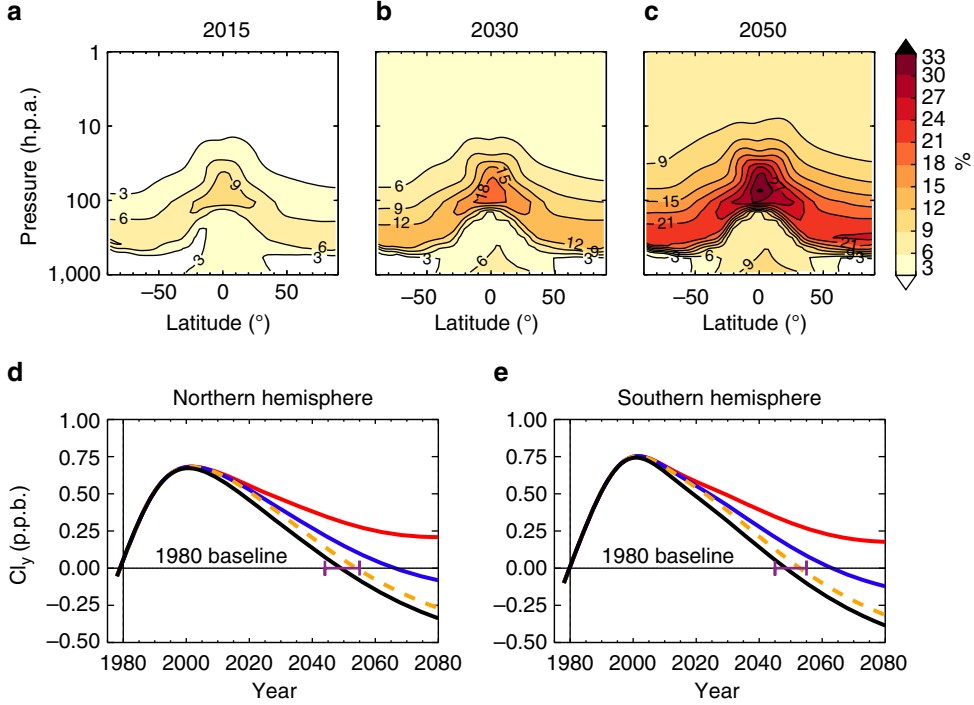

**Figure 3 | Contribution of $CH_2Cl_2$ to stratospheric inorganic chlorine and changes in total inorganic chlorine relative to 1980 baseline.** (**a–c**) Percentage (%) of total stratospheric inorganic chlorine ($Cl_y$) derived from $CH_2Cl_2$ in 2015, 2030 and 2050 under model run Scenario 1 (assuming $CH_2Cl_2$ continues to increase at the mean rate observed over the 2004–2014 period). (**d,e**) Modelled annual mean mid-latitude $Cl_y$ change in northern (35–60° N) and southern (35–60° S) hemisphere. The $Cl_y$ change is expressed in parts per billion (p.p.b.) relative to 1980 baseline at 50 hPa (lower stratosphere). $Cl_y$ changes are shown for model simulations without $CH_2Cl_2$ (black) and with $CH_2Cl_2$ under growth Scenarios 1 (blue, see above) and 2 (red, assuming $CH_2Cl_2$ increases at the mean rate observed over the 2012–2014 period), and for the no additional growth Scenario 3 (orange). The projected dates when $Cl_y$ returns to 1980 levels in the NH are 2050 (no $CH_2Cl_2$), 2067 (Scenario 1) and 2054 (Scenario 3). In the SH: 2049 (no $CH_2Cl_2$), 2064 (Scenario 1) and 2053 (Scenario 3). The horizontal purple lines show best estimated range of 1980 return dates from previous CCMs[6] which did not include $CH_2Cl_2$.

stratosphere, $Cl_y$ return dates occur later than in the lower stratosphere and are less sensitive to future $CH_2Cl_2$ growth (Supplementary Fig. 1). Here, the contribution of $CH_2Cl_2$ to total $Cl_y$ remains at <10% in 2050 (Fig. 3c).

**Impact of past and potential future $CH_2Cl_2$ growth on ozone.** We consider next the impact of $CH_2Cl_2$ on ozone. Ozone is most sensitive to $CH_2Cl_2$ in polar regions; by the time air reaches high latitudes all chlorine that entered the stratosphere as $CH_2Cl_2$ has been converted to $Cl_y$. The largest ozone decreases attributable to $CH_2Cl_2$ are simulated in the SH, where the Antarctic Ozone Hole—the most drastic manifestation of the effect of halogen-driven ozone loss—forms each spring. Although modest, the impact of $CH_2Cl_2$ is non-negligible in the present day with springtime column ozone up to ~3%, or 6 dobson units (DU), lower in simulations in which $CH_2Cl_2$ is considered relative to an atmosphere without $CH_2Cl_2$ in 2016 (Supplementary Fig. 2). The equivalent relative ozone decrease in the spring of 2010 was ~1.5% (3 DU), which allows quantification of $CH_2Cl_2$ increases on polar ozone depletion during spring, over this period, and highlights that this impact has already doubled in the past 6 years alone.

Figure 4 shows simulated ozone changes due to $CH_2Cl_2$ in the recent past and future period. Ozone loss due to further $CH_2Cl_2$ growth is projected to increase significantly in coming decades. By 2050, expressed as an annual mean, ozone is ~6% lower in the Antarctic lower stratosphere under growth Scenario 1 (Fig. 4a), and annual mean column ozone is decreased by up to 8 DU, relative to the no $CH_2Cl_2$ simulation (Fig. 4c), against a

background of recovering ozone. At mid-latitudes, column ozone decreases are smaller (up to several DU) and despite $CH_2Cl_2$ making a relatively large contribution to $Cl_y$ in the tropics, ozone decreases here remain small (<1%). Recall, our simulations in the future period reflect the ozone response to changes in projected stratospheric composition only, comparing scenarios with $CH_2Cl_2$ growth to one with no $CH_2Cl_2$. Future ozone is also expected to be influenced by other factors, including climate-driven changes to stratospheric temperature and circulation[7,12,25] and possibly due to changes in natural halocarbon emissions[26]. By isolating the ozone response due to $CH_2Cl_2$, we highlight its increasing influence on ozone evolution in coming decades. Such findings could also be relevant from a climate perspective[21] as ozone absorbs both ultraviolet and infrared radiation, and as ozone perturbations in the lower stratosphere cause a relatively large radiative effect[27].

Although the severity of ozone loss over Antarctica is most strongly affected by the abundance of reactive halogens, inter-annual variability in temperature and dynamical influences that determine strength of the polar vortex provide additional influence. For example, relatively large levels of springtime (September–November) ozone have been observed (that is, less loss) following stratospheric warming events, such as in 2002 (ref. 28), during which the occurrence of polar stratospheric clouds was relatively low[29]. The sensitivity of the SH column ozone decrease due to $CH_2Cl_2$ to a range of assumed stratospheric meteorology in the future period (see Methods) is shown in Fig. 5. In all cases, the impact of $CH_2Cl_2$ is substantial, with the greatest ozone decreases towards the end of the century predicted under the assumed 2012 (base meteorology) and 2006 (relatively cold

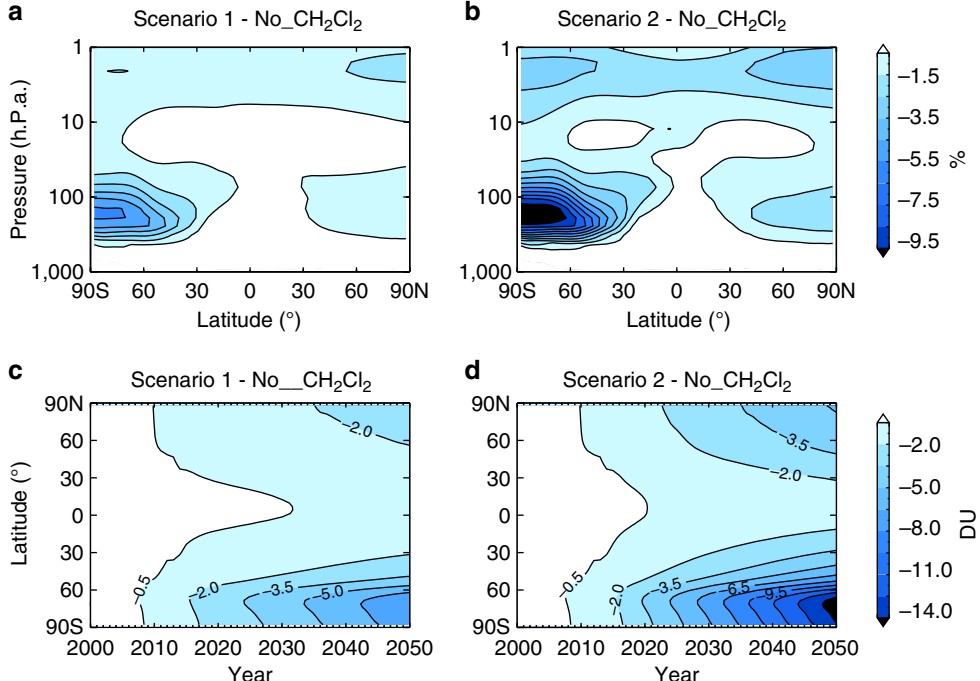

**Figure 4 | Stratospheric ozone decreases due to CH$_2$Cl$_2$.** (**a**) Difference in zonal mean annual mean ozone (%) between run Scenario 1, assuming surface CH$_2$Cl$_2$ continues to increase at the mean rate observed over the 2004–2014 period, and run no_CH$_2$Cl$_2$ in 2050. (**b**) As **a** for Scenario 2, assuming CH$_2$Cl$_2$ continues to increase at the mean rate observed over the 2012–2014 period. (**c**) Difference in zonal mean annual mean column ozone (DU) between run Scenario 1 and run no_CH$_2$Cl$_2$ as a function of year. (**d**) As **c** for Scenario 2.

Antarctic winter) conditions. However, note, these CTM simulations did not explicitly consider climate-driven cooling of the upper stratosphere or changes in stratospheric dynamics (see below).

## Discussion

The Montreal Protocol has been extremely successful in alleviating polar ozone loss. For example, it has been estimated the Antarctic Ozone Hole would have been 40% larger by 2013 had the Protocol not come into effect[30]. Based on current understanding, the Ozone Hole is expected to recover in this century. The 17 chemistry-climate models (CCMs) that took part in the Stratospheric Processes and their Role in Climate (SPARC) CCMVal project predict that the Antarctic Ozone Hole—defined by the October average column ozone abundance—will return to pre-1980 levels between 2046 and 2057 (ref. 7), in line with the anticipated decline in controlled ozone-depleting substances. More recent model simulations have suggested a later ozone return date that approaches the end of the century[31,32], though none of these previous estimates have considered CH$_2$Cl$_2$, or indeed CH$_2$Cl$_2$ growth.

Figure 6 shows our model estimate of Antarctic Cl$_y$ and column ozone evolution with and without CH$_2$Cl$_2$. Antarctic column ozone returns to pre-1980 levels in 2065 when CH$_2$Cl$_2$ is not considered. The inclusion of CH$_2$Cl$_2$ delays the ozone return by 30 years under growth Scenario 1 (note, under Scenario 2 ozone does not return to pre-1980 levels this century). This is the difference that would be expected between an atmosphere without any CH$_2$Cl$_2$, for example, if its production was completely phased out, to an atmosphere in which sustained future growth of CH$_2$Cl$_2$ continues. Recall, Scenario 1 simply assumes CH$_2$Cl$_2$ continues to increase in the atmosphere following the average upward trend observed over the last

decade. We estimate that even if the stratospheric input of CH$_2$Cl$_2$ remains at 2016 levels (that is, Scenario 3, a no future growth scenario), the return of ozone in the Antarctic Ozone Hole region is still delayed by $\sim$5 years, relative to the no CH$_2$Cl$_2$ simulation.

The above return dates are calculated from a 1980 ozone baseline in an atmosphere that is assumed to contain no CH$_2$Cl$_2$. Emissions of CH$_2$Cl$_2$ in 1980 are highly uncertain, owing to a paucity of atmospheric measurements at that time. However, a global emission source of $\sim$800 Gg CH$_2$Cl$_2$ per year in the early 1980s has been derived from bottom–up methods[33]. Using this estimate and adjusting the ozone baseline to account for the presence of CH$_2$Cl$_2$ would change the 1980 return dates to 2063 (no CH$_2$Cl$_2$ simulation) and 2090 (growth Scenario 1). Clearly, additional increases in atmospheric CH$_2$Cl$_2$ could lead to an underestimate of the timescale for stratospheric ozone recovery in coming decades. Furthermore, as CH$_2$Cl$_2$ already has a non-negligible impact on polar ozone, which has increased in the recent past (Supplementary Fig. 2), in the nearer term CH$_2$Cl$_2$ growth could confound the search for ozone recovery attributable to the phase-out of controlled ozone-depleting compounds. We note that as the Antarctic column ozone changes scale to a good approximation linearly with the Cl$_y$ load under each CH$_2$Cl$_2$ scenario (Supplementary Fig. 3), our results can be used to estimate the impact of different future trajectories of CH$_2$Cl$_2$, or indeed other chlorinated VSLS, on ozone.

While the future trajectory of CH$_2$Cl$_2$ is uncertain, in the absence of policy controls on its production, it is likely that future trends in this gas will fall within the range of the three scenarios presented here. From Fig. 5, the absolute impact of CH$_2$Cl$_2$ on SH polar ozone will exhibit some year-to-year variability, depending on stratospheric meteorology (Fig. 5), though the 1980 Antarctic ozone return date delay due to CH$_2$Cl$_2$ exhibited a weak sensitivity to the choice of future meteorology in our SLIMCAT simulations. However, we note these runs did not consider

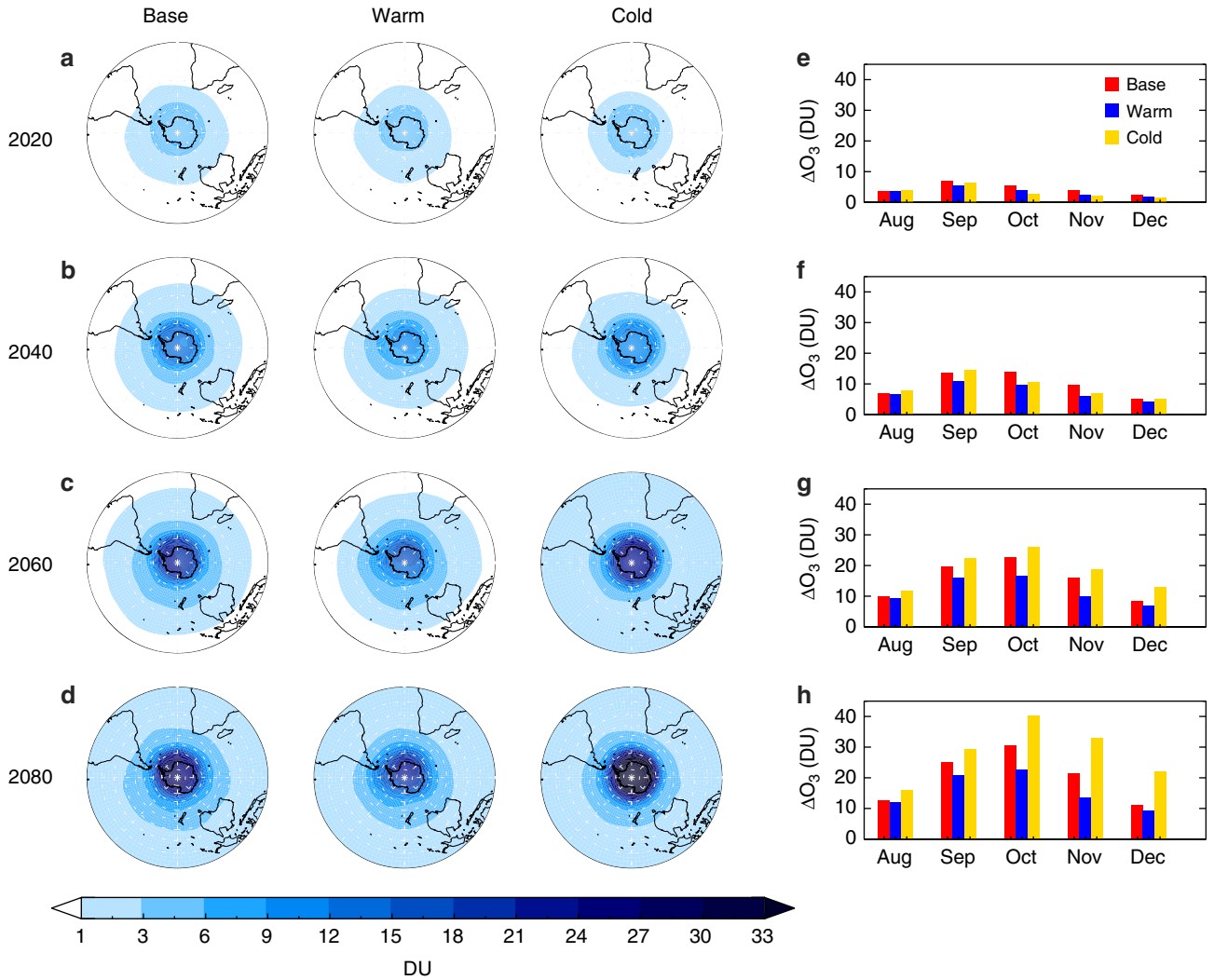

**Figure 5 | Sensitivity of SH ozone loss due to CH$_2$Cl$_2$ to different stratospheric meteorology.** (**a–d**) Spring-time mean column ozone decrease (DU) in SLIMCAT in 2020, 2040, 2060 and 2080 calculated as run no_CH$_2$Cl$_2$ minus run Scenario 1 (assuming surface CH$_2$Cl$_2$ continues to increase at the mean rate observed over the 2004–2014 period). Shown are results for simulations assuming either 2012 (base meteorology), or 2002 (relatively warm) or 2006 (relatively cold) meteorological conditions in the future. (**e–h**) Corresponding monthly mean column ozone decreases south of −60° latitude.

inter-annual dynamical variability in the future or the effects of climate change, such as an acceleration of the Brewer-Dobson circulation or prolonged stratospheric cooling, as predicted by most CCMs. Although ozone trends in the Antarctic lower stratosphere are most strongly affected by trends in ozone-depleting gases (that is, the time evolution of the stratospheric halogen content) and relative to the Arctic, are less sensitive to anticipated greenhouse gas changes in the future[7], climate change is expected to accelerate ozone recovery in this region[12]. Furthermore, given the current spread of return dates predicted by CCMs, the delay in the ozone return due to CH$_2$Cl$_2$ is likely to vary between models, even when considering the same CH$_2$Cl$_2$ growth scenarios. For a given 1980 ozone baseline, models with an inherent slower rate of ozone return will see a greater impact of CH$_2$Cl$_2$ on the ozone return date. This is due to the divergence of stratospheric Cl$_y$ concentrations in time between an atmosphere without CH$_2$Cl$_2$ in the future, and one in which sustained CH$_2$Cl$_2$ growth occurs.

To explore the above, we performed simulations with the UMSLIMCAT CCM, forced by the Intergovernmental Panel on Climate Change (IPCC) RCP (Representative Concentration Pathway) 6.0 scenario, and with evolving meteorology in the

future. UMSLIMCAT was evaluated in the SPARC CCMVal project[12] and is participating in the ongoing Chemistry-Climate Model Initiative[34] (CCMI – Methods section). Our CCM results show marked differences between a run with CH$_2$Cl$_2$ growth (Scenario 1) and a reference run without CH$_2$Cl$_2$ (Fig. 7a). The largest ozone decreases occur over Antarctica, where (i) the difference in October mean ozone between the with and without CH$_2$Cl$_2$ runs is statistically significant at the 95% confidence interval, and (ii) the influence of CH$_2$Cl$_2$ on ozone increases in coming decades—corroborating our main CTM findings. In both models, inclusion of CH$_2$Cl$_2$ growth delays the return of Antarctic lower stratospheric Cl$_y$ to 1980 levels by 13 years. In UMSLIMCAT, column ozone return is delayed by 17 years (Fig. 7b). This is a substantial delay, albeit it is smaller than that predicted by the CTM, which assumed fixed dynamics in the future, and which predicts more severe springtime polar ozone loss (that is, for a given amount of chlorine from CH$_2$Cl$_2$, the ozone impact is smaller in the CCM). Ozone increases as a result of upper stratospheric cooling, a consequence of climate change[7], may provide additional influence on the magnitude of the ozone delay from CH$_2$Cl$_2$ in CCM experiments. However, our CCM shows that ozone closely tracks the trajectory of Cl$_y$, highlighting

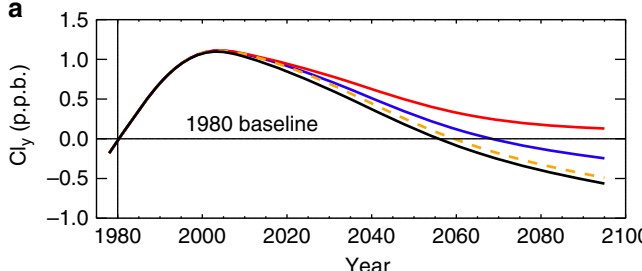

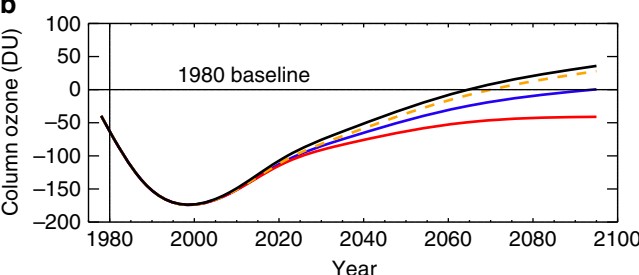

**Figure 6 | Long-term changes in stratospheric inorganic chlorine and column ozone over Antarctica.** (**a**) Modelled October mean change in inorganic chlorine ($Cl_y$) in the lower stratosphere (50 hPa) over Antarctica (60–90° S). $Cl_y$ is expressed in parts per billion (p.p.b.) relative to a 1980 baseline. (**b**) Corresponding modelled October mean change in Antarctic stratospheric column ozone (DU) relative to 1980. $Cl_y$ and ozone changes shown for SLIMCAT model simulations without $CH_2Cl_2$ (black) and for $CH_2Cl_2$ Scenario 1 (blue, surface $CH_2Cl_2$ continues to increase at the mean rate observed over the 2004–2014 period), Scenario 2 (red, surface $CH_2Cl_2$ continues to increase at the mean rate observed over the 2012–2014 period) and Scenario 3 (orange, no future growth). Note, 1980 baseline is calculated from a model simulation performed with 2012 meteorology, in a similar manner to the forward simulations, to isolate the impact of $CH_2Cl_2$ growth from inter-annual variability due to meteorology. For $Cl_y$, the calculated return dates with respect to this baseline are 2056 (no $CH_2Cl_2$), 2069 (Scenario 1) and 2060 (Scenario 3). Similarly for ozone, 2065 (no $CH_2Cl_2$), 2095 (Scenario 1) and 2071 (Scenario 3).

the dominant influence of the halogen loading on Antarctic ozone trends[7,12]. As previously noted, the delay from $CH_2Cl_2$ growth will vary across models and crucially these results highlight that $CH_2Cl_2$ growth should be considered within the framework of multi-model ozone assessments in the future. As the Antarctic Ozone Hole is known to affect surface climate of the SH in several ways, such as by modifying the Southern Annular Mode[35], a delayed recovery caused by $CH_2Cl_2$ may also be relevant for refining future climate predictions.

In summary, although currently modest, the impact of $CH_2Cl_2$ on stratospheric ozone is increasing and if $CH_2Cl_2$ concentrations continue to increase they could significantly offset a portion of the decline in anthropogenic chlorine provided by the Montreal Protocol. This adds uncertainty into future assessments of ozone evolution and could lead to a significant delay in recovery of the ozone layer, particularly over Antarctica. Finally, we note that while this study has focused on $CH_2Cl_2$, for which long-term surface monitoring data exists, several other anthropogenic VSLS (for example, 1,2-Dichloroethane, $C_2H_4Cl_2$) have been detected in Earth's atmosphere[22], though atmospheric measurements of these compounds are sparse. A broader consideration of the atmospheric trends of these non-Montreal Protocol compounds and other VSLS would be beneficial to improve future ozone predictions.

## Methods

**Surface $CH_2Cl_2$ measurements.** Surface measurements of $CH_2Cl_2$ from NOAA's long-term monitoring network[36] were analysed. Results from paired flask samples collected at 13 sites were used to derive surface $CH_2Cl_2$ mixing ratios over the 2004–2014 period. The observed data were averaged in 5 latitude bins (60–90° N, 30–60° N, 0–30° N, 0–30° S and 30–90° S) and trends over the above period were derived in each bin (selected bin results are shown in Fig. 1a). These data form the basis of our current emission estimates and future $CH_2Cl_2$ growth scenarios.

**Aircraft $CH_2Cl_2$ measurements.** Aircraft measurements of $CH_2Cl_2$ around the tropopause (Fig. 1d, inset) were obtained onboard the Global Hawk unmanned aircraft deployed during the 2011 and 2013 legs of the National Aeronautics and Space Administration (NASA) Airborne Tropical Tropopause Experiment (ATTREX) mission[18]. $CH_2Cl_2$ mixing ratios were derived by the University of Miami from whole air samples analysed using gas chromatography/mass spectrometry.

**Future $CH_2Cl_2$ growth scenarios.** Two scenarios describing the future evolution of surface $CH_2Cl_2$ were derived by extrapolating observed surface $CH_2Cl_2$ trends in 5 latitude bins (see above). The forward scenarios cover the period 2015–2100. Scenario 1 assumes that surface $CH_2Cl_2$ increases linearly over this period, with a growth rate equal to the mean growth rate observed over the 2004–2014 period. In NH mid-latitudes, surface $CH_2Cl_2$ increases at a rate of 2.85 p.p.t. per year under Scenario 1 (Fig. 1c). Scenario 2 is a larger growth scenario and assumes that $CH_2Cl_2$ increases linearly with a rate equal to that observed over the 2012–2014 period only. In these years, $CH_2Cl_2$ growth was large compared to the decadal average used for Scenario 1, resulting in a surface trend of 6.1 p.p.t. per year in NH mid-latitudes. A third scenario, Scenario 3, was also considered in which it is assumed that no further growth of $CH_2Cl_2$ occurs beyond 2016 (that is, the surface concentration of $CH_2Cl_2$ is fixed throughout the forward simulation at 2016 levels, Fig. 1).

**Global emission estimates of $CH_2Cl_2$.** A simple 1-box model, previously used to study $CH_4$ and $CH_3CCl_3$ emissions, was used to derive global $CH_2Cl_2$ emissions required to sustain the observed surface concentrations. The model treated emissions of $CH_2Cl_2$ with a parameterized global mean lifetime of 0.43 years, based on this work. In the present day, global $CH_2Cl_2$ emissions estimates (Fig. 2) agree well with similar independent estimates[15]. The box model was also used to estimate that an emission source of ∼2.8 Tg $CH_2Cl_2$ per year is required to sustain the modelled 2050 atmospheric $CH_2Cl_2$ concentration under Scenario 1.

**Chemical transport model and experiment design.** The TOMCAT/SLIMCAT global three-dimensional CTM[37] was used to calculate the sensitivity of stratospheric chlorine and ozone to future growth in atmospheric $CH_2Cl_2$. The model has been widely evaluated and used in previous studies of tropospheric and stratospheric chemistry. The model is forced with meteorological fields from the European Centre for Medium-Range Weather Forecasts (ECMWF) ERA-Interim reanalysis data set. We first derived the time-dependent stratospheric injection of $CH_2Cl_2$ over the 2004–2100 period using a detailed tropospheric configuration of TOMCAT. The model contains a comprehensive description of tropospheric chemistry, including VSLS oxidation, and reproduces the tropospheric abundance of chlorine-containing VSLS well in the present day[22]. A transient simulation between 2004 and 2014 was performed in which surface $CH_2Cl_2$ was constrained in the model using the time-dependent surface measurements from NOAA, in the 5 latitude bins discussed above. Between 2015 and 2100, the model was integrated twice using the projected surface $CH_2Cl_2$ loadings from Scenario 1 and Scenario 2. We assumed present day meteorology and emissions of precursor gases that control tropospheric chemistry in the future period. The stratospheric chlorine injection from $CH_2Cl_2$ over these periods is given in Fig. 1d.

We next performed three simulations using a detailed stratospheric model configuration of TOMCAT/SLIMCAT, containing a comprehensive stratospheric chemistry scheme that includes a full description of processes relevant to polar ozone depletion. This version of the model reproduces observed stratospheric ozone trends well[21,30]. Experiments were performed over the 1980–2100 period at a horizontal resolution of 2.8° × 2.8° and with 32 vertical levels from the surface to ∼60 km. The stratospheric $CH_2Cl_2$ loading was prescribed in experiments Scenario 1 and Scenario 2 using the time-dependent loadings derived above. Annual averages of these data are given in Supplementary Tables 1 and 2. $CH_2Cl_2$ was only considered from 2004 onwards and the two growth scenarios diverge from 2015. A reference simulation in which $CH_2Cl_2$ was not considered (experiment No_$CH_2Cl_2$) was also performed. Time-dependent surface mixing ratios of all other long-lived ozone-depleting compounds (for example, CFCs, HCFCs, halons and others), $CH_4$ and $N_2O$, were taken from the World Meteorological Organization (WMO) A1 Scenario[7,15]. Our approach is to examine the sensitivity of future $Cl_y$ and ozone to increasing $CH_2Cl_2$. Therefore, in the future period annually repeating meteorology was assumed for the arbitrarily chosen year 2012 (referred to here as 'base meteorology'). We also examined the sensitivity of ozone changes due to $CH_2Cl_2$ growth to different and more remarkable assumed years of meteorology in the future period: 2002 (a year of relatively warm polar stratospheric temperatures) and 2006 (a year of very cold temperatures).

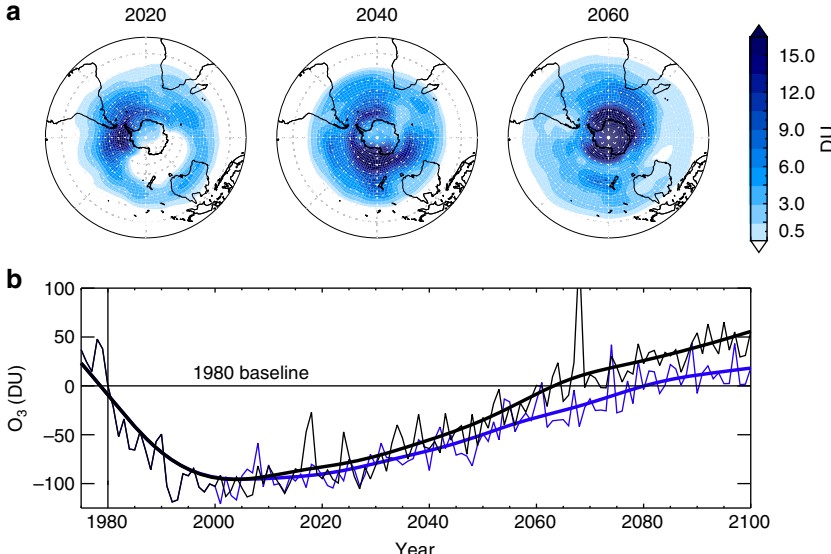

**Figure 7 | Future impact of CH$_2$Cl$_2$ growth on Antarctic column ozone and ozone trend from CCM simulations.** (**a**) Springtime mean column ozone decrease (DU) due to CH$_2$Cl$_2$ in the SH. (**b**) October mean Antarctic stratospheric ozone column (DU) relative to 1980. Results are shown for UMSLIMCAT run with CH$_2$Cl$_2$ growth Scenario 1 (blue, surface CH$_2$Cl$_2$ continues to increase at the mean rate observed over the 2004–2014 period) and without CH$_2$Cl$_2$ (black). While inter-annual variability is large, the two ozone time series are statistically different at the 95% significance level according to a Student's $t$-test ($P$ value = 0.02). Ozone returns to the 1980 baseline in the year 2064 (without CH$_2$Cl$_2$) and in 2081 (Scenario 1).

**Chemistry-climate model experiments.** UMSLIMCAT is a global CCM that has been evaluated extensively within the recent CCMVal and CCMI multi-model inter-comparison initiatives[12,34]. It is based on the troposphere–stratosphere–mesosphere version of the Met Office Unified Model (UM v4.5). The model has 64 vertical levels extending from the surface to 0.01 hPa (~80 km), and was here run at a horizontal resolution of 3.75° × 2.5° (ref. 38). The stratospheric chemistry scheme is similar to that of the SLIMCAT CTM (see above). We performed a transient control simulation without CH$_2$Cl$_2$ over the 1970 to 2100 period. The loading of long-lived ozone-depleting compounds and greenhouse gases was prescribed according to the IPCC RCP 6.0 scenario. A similar run but including time-varying CH$_2$Cl$_2$ according to growth Scenario 1 was also performed.

**Estimates of stratospheric chlorine and ozone return dates.** Estimates of stratospheric Cl$_y$ and ozone return dates relative to a 1980 baseline are presented. For the CTM experiments, this baseline was calculated from a 1980 model simulation run with 2012 meteorology (consistent with our forward simulations), to isolate the effect of changing composition on the return dates. Estimated Cl$_y$ return date ranges are also shown (purple lines in Fig. 3) from previous CCM simulations[7,12] that did not consider CH$_2$Cl$_2$. The model time series shown in Fig. 3 and all subsequent figures were smoothed by applying a 1:2:1 filter iteratively 30 times, consistent with previous studies[12].

**Code availability.** The TOMCAT/SLIMCAT model is supported by the Natural Environment Research Council (NERC) and the National Centre for Atmospheric Science (NCAS) and is available to UK academic institutions working with these organizations. Enquiries about the model code should be directed to M.P.C. Post processing of model output was performed using Interactive Data Language (IDL) and the code is available on request from the corresponding author (R.H.). Output from all model simulations is available on request from the corresponding author (R.H.).

**Data availability.** The NOAA surface CH$_2$Cl$_2$ data are available to download at the following web address: http://www.esrl.noaa.gov/gmd/dv/ftpdata.html. Use of the data in a presentation, publication or report requires that users contact the principal investigator (S.A.M.) first to discuss your interests. The NASA CH$_2$Cl$_2$ data from the ATTREX missions are publically available at the following web address: https://espoarchive.nasa.gov/.

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

## Acknowledgements

R.H. thanks the Natural Environment Research Council (NERC) for funding a research fellowship (NE/N014375/1). We thank the European Research Council (ERC) for support under the ACCI project (Project number 267760). M.P.C. thanks the Royal Society for a Wolfson Research Merit Award. We thank Wuhu Feng (NCAS) for help with SLIMCAT. This work was performed using the Archer and ARC2 high performance computing facilities. NOAA measurements were supported in part by the NOAA Climate Program Office's AC4 programme. We thank the two anonymous reviewers and Dr Robyn Schofield whose constructive comments strengthened the manuscript.

## Author contributions

R.H., M.P.C. and S.S.D. conceived the idea and initiated the study in collaboration with J.A.P. and S.A.M.; R.H., M.P.C. and A.A.L. performed and analysed the SLIMCAT model runs. S.S.D. performed the UMSLIMCAT runs. The $CH_2Cl_2$ data and growth scenarios are based on ground-based measurements obtained by S.A.M.; R.H. and A.A.L. prepared the figures. All authors discussed the results and commented on the manuscript.

## Additional information

**Competing interests:** The authors declare no competing financial interests.

**Publisher's note**: 

