## [Peer Review File · Nature Communications]

Reviewers' comments:

Reviewer #1 (Remarks to the Author):

This is an excellent paper that is highly appropriate for publication in Nature Communications. I have only one so-called Major Comment.

Major comment:

1. Given the journal, the paper would be *much stronger* if a new paragraph could be added, focusing on the industrial applications of dichloromethane that are likely responsible for the rise in atmospheric abundance. Please add upon revision, assuming this paper goes forward: the co-author team has the expertise to write this into the paper, rather than to leave to others asked to comment on the paper.

Minor comments, including a few silly word-smithing suggestions:

1. Line 14: "As a consequence" would read better to most

2. Line 32: citations are overall excellent IMHO but for the citation to "ozone recovery is underway", should consider adding a citation to Yang et al., JGR, 2008:

<http://onlinelibrary.wiley.com/doi/10.1029/2007JD009675/full>

in addition to the other papers that are cited.

3. Line 49: strike first use of "growth" ... don't need to use this word twice

4. Line 56: suggest "is" rather than "was"

5. Line 68: can strike "measurement"

6. Line 71: To readers from outside the field, which Nature Communications aspires to reach, it will seem puzzling that the emissions of CH₂Cl₂ can be so much larger than the emissions of CFCs and other ozone-depleting gases, without CH₂Cl₂ being the dominant ODS. Of course, it is the short lifetime of CH₂Cl₂ that accounts for this so-called puzzle.

I suggest adding a new paragraph between lines 71 and 72, or else a well crafted sentence at the end of "gases peaked", to explain this issue: i.e., try to anticipate some Chemical Engineers in the manufacturing industry struggling with this concept.

7. Lines 76 and 79: it cannot be possible that the growth rate of Scenario 1 is 2.85 ppt/year and 4.49 %/year. This can only be true for one year. Same for 6.1 ppt/year and 8.56 %/year for Scenario 2.

Also, one of these numbers make sense given Table S1, where for Scenario 1 between 2080 and 2079, we see a change of 140.1 – 138.4 = 1.7 ppt, and for Scenario 2 we see a change of 300.0 – 296.0 = 4 ppt/year.

What pray tell is the assumption for the numerical rise of each scenario? Please get this straight in the paper, and present either as ppt/year or %/year, but not as both.

Finally, Table S2 should extend to an end year consistent with the figures. The table as submitted stops in 2080, but the figures extend past 2080.

8. Lines 93 and 94: I suggest "By 2050 under Scenario 2, ..."

9. Line 100: suggest a new paragraph at "Under Scenario 2", or else somewhere in this 21 line paragraph, because the paper will read better with shorter paragraphs.

10. Line 103: May want to consider adding a citation and a phrase to a paper recently published by Oman et al. in GRL:

<http://onlinelibrary.wiley.com/doi/10.1002/2016GL070471/full>

as well as the Yang et al. paper in ACP on which one of the co-authors participated:

<http://www.atmos-chem-phys.net/14/10431/2014>

11. Line 119: I suggest a new paragraph at "Figure 4 shows" and a sentence describing what is depicted in the figure, before the figure is interpreted

12. Line 134: can strike "already" ... word is not needed and detracts from the message

13. Line 142: might consider citing Oman et al., GRL, 2016 along with ref #27

14. Lines 149 and 152: paper will read better using "growth of CH₂Cl₂"

15. What are the red and blue circles with error bars, on Figure 1d? These do not seem to be explained.

16. The large headers on the figures will detract from the conveyance of the info in the figures, upon publication. See if these labels can be placed inside the panels of the figures, and try to reduce white space between panels.

End of review ... thanks for submitting such an easy to review paper, on a Saturday night.

Reviewer #2 (Remarks to the Author):

In this manuscript Hossaini and colleagues consider the recently-published increases in atmospheric dichloromethane, CH₂Cl₂, mixing ratios and create future scenarios that outline the potentially substantial impacts of continued increases in CH₂Cl₂ emissions on stratospheric ozone levels and Antarctic ozone hole recovery.

The paper is compact and well written, with well-presented figures. I am happy with the quality of the original data. Figures are easy to understand and the supplementary information is appropriate. However, I feel this paper fails to provide a considerable advance to our understanding of this field.

The first result presented by the authors is the recent growth in atmospheric CH₂Cl₂ (lines 62-66). This increase has already been characterised and published^{1,2,3}, including in the author's prior work. Their estimated global emission of 1 Tg CH₂Cl₂/year (line 69) is also similar to the NOAA estimate reported in the last WMO report⁴ of 841 ±183 Gg CH₂Cl₂/year. The similarity is possibly linked to recent period of slow CH₂Cl₂ growth discussed below.

The authors then go on to present two future emission scenarios, both based on "...the extrapolation of observed long-term surface trends" (lines 73-74). Scenario 1 is the more conservative, assuming CH₂Cl₂ continues to increase at the mean rate it has done 2004-2014. Scenario 2 is more extreme, considering a projected increase based on the fast growth observed between 2012 and 2014. I would contest that, considering: (1) CH₂Cl₂ emissions vary considerably year on year (Fig. 1b and line 65) and (2) that in the past three years or so the CH₂Cl₂ growth rate has slowed (Fig. 1a), both scenarios could be over estimations and the authors should address this uncertainty and potentially revise these scenarios accordingly. I draw particular attention to point (2) and the slowing of the growth rate in Fig. 1a since 2013. Data in this figure come from NOAA measurements and a recent (31st Oct 2016) check of the NOAA data freely available on <http://www.esrl.noaa.gov/gmd/hats/gases/CH2Cl2.html> suggests that this plateau in atmospheric surface CH₂Cl₂ concentrations is continuing. The average CH₂Cl₂ concentration for the first half of 2016 for the five stations between 30-60 °N (to fit with Fig 1a) is ~64 ppt⁵, similar to results from 2013-2015. To ignore this three to four year pattern when creating future projections is potentially misleading.

This is part of an important fact about CH₂Cl₂ and other VSLC Cl species: these species have short (~5 months for CH₂Cl₂⁴ 4) lifetimes and so changes in emissions will have relatively rapid effects on atmospheric abundance. Unlike CFCs, which the authors make comparisons to (e.g. line 70), the tropospheric chlorine burden created by a few years of strong emissions and growth can, relatively quickly, be reversed. This has two impacts for this work:

- As discussed above, both Scenario 1 and Scenario 2 are inferred from current situations that may already be changing.
- Unlike longer-lived gases, where changes in sinks could impact recovery back to pre-industrial atmospheric concentrations, the driving force in future atmospheric CH₂Cl₂ concentrations will be industrial emissions, a factor the authors admit are "poorly characterized" (line 52). I feel that the two limited scenarios discussed by the authors are inadequate for covering the potential fluctuations in anthropogenic supply and demand, especially considering the variations in growth rate shown over the past decade (Fig. 1b).

The authors next go on to discuss model results showing that increasing CH₂Cl₂ emissions into the stratosphere based on increasing surface concentrations are relatively simple. A 'back of the envelope' calculation where I took the average NOAA 2016 CH₂Cl₂ mixing ratio of 64 ppt and projected this forward to 2050 with an increase of 2.85 ppt/year (manuscript Scenario 1, lines 75-76) and then reduced the 2050 value by 18% (representing the 18% decrease in [CH₂Cl₂] observed between the marine boundary layer and the layer of zero radiative heating given in both the 2010 and 2014 WMO reports^{4,6}) gave me a rough value of ~133 ppt CH₂Cl₂ entering the stratosphere in 2050. This is around their lower estimation (Fig. 1d). To use this paper to move this research field on from the previous observational work discussed previously^{1,2,3} it would be nice to see the scenarios used to constrain limiting factors we are not able to easily calculate without a model. For example, the authors mention the potential impact of other physical and chemical changes on stratospheric Cl_y^{VSLC} and ozone depletion, such as climate-driven changes to stratospheric temperature and circulation (lines 127-128) and tropospheric-stratospheric exchange. The impact of changes in these areas need to be considered when making future predictions of the kind discussed here.

In summary, the authors present previously published data combined with somewhat-limited model projections. Whilst the current and future significance of non-Montreal Protocol regulated VSLC Cl species, such as CH₂Cl₂, is an important topic for debate I

feel that this paper does not do enough to address the key limitations in our current understanding of this topic.

Reviewer #3 (Remarks to the Author):

Review of The increasing threat to stratospheric ozone from dichloromethane by Hossaini et al.,

This manuscript represents a thorough and vital analysis of an emerging atmospheric trace gas species of concern that has large implications for stratospheric ozone recovery. To date this analysis has been missing from the literature. Very short-lived substances are not regulated under the current Montreal protocol framework for the protection of stratospheric ozone. Hence, the findings that a delay in ozone recovery of 30 years if the growth rate continues as present is of relevance to a wide community. In a more aggressive growth scenario, ozone recovery in the 21st century is precluded. With approximately 50% of the climate change experienced in spring and summer in the Southern hemisphere over the past 30 years attributed to the springtime Antarctic Ozone hole, a delay in stratospheric ozone recovery has major implications for the climate system. Therefore, this research has broad readership from policy, climate, atmospheric chemistry, and epidemiology interests and I recommend its publication in Nature communications. I have some questions and suggestions that I list below.

The title could have more impact representing the implications of this work - delay in ozone recovery could be included.

The writing in places is a little awkward - 'In consequence' is not an accessible construct - I suggest changing both instances (lines 14 and 35).

Line 42: qualify with stratosphere 'minor role in stratospheric ozone...'

More examples of the diverse uses of dichloromethane, and perspective on alternatives would add to the introduction (paragraph beginning line 37) - i.e. why has the growth occurred in this species?

Line 44 suggest replacing 'due to expected relatively ' with 'therefore'

Line 67 it could be interesting to note here (like most CFCs), while the emissions are weighted toward the Northern hemisphere, the consequences are disproportionately experienced in the Southern hemisphere.

Line 79 - sentence The 2012-2014 could be removed - this point has been made in the previous sentence.

The point that your model accurately simulates both the surface and the UTLS CH₂Cl₂ observations providing confidence that the emissions and lifetimes are well represented needs to be made more clearly (paragraph beginning line 72).

Line 89 write the stratospheric lifetime here (and reference)

Paragraph beginning line 87 - what is the role of chlorine introduced to the lower stratosphere under high volcanic aerosol loading? If you were able to quantify this with an additional scenario run it would add to the future possible implications of this work.

Line 132. The climate effect of the Antarctic ozone hole is outlined in chapter 4 of the 2014 WMO

report - the text around this could be tightened significantly.

Line 161 Note that under growth scenario 2 no recovery is seen by 2100

Line 167 the strengthening of the Brewer Dobson circulation is also not included.

Line 213 include the reference for 0.43 year lifetime.

Line 217 what is the uncertainty introduced in your methodology by using modern day meteorology? It should be possible to quantify this by having simulation with GCM evolving meteorology including a stronger BD circulation, and cooler stratospheric conditions.

Robyn Schofield

Response to Reviewer #1

We thank the reviewer for his/her comments. These are reproduced below in **bold**, followed by our responses in *red italics*.

Reviewer #1 (Remarks to the Author):

This is an excellent paper that it highly appropriate for publication in Nature Communications.

We thank the reviewer for these very supportive comments.

I have only one so-called Major Comment.

Major Comment:

1. Given the journal, the paper would be *much stronger* is a new paragraph could be added, focusing on the industrial applications of dichloromethane that are likely responsible for the rise in atmospheric abundance. Please add upon revision, assuming this paper goes forward: the co-author team has the expertise to write this into the paper, rather than to leave to others asked to comment on the paper.

OK, we agree that such a paragraph would strengthen the discussion. Although the precise nature of the CH₂Cl₂ source responsible for the observed upward trend is unknown (it could be due to several applications of CH₂Cl₂), we have added the following text to the revised manuscript (see first paragraph of Results).

“At present, it is unknown if a single industrial application of CH₂Cl₂, or several, is contributing to the observed upward trend. As a common solvent, CH₂Cl₂ has numerous applications, which include use in metal cleaning/degreasing, in paint remover, and use by the pharmaceutical industry for preparing drugs. It is also used as blowing agent in production of foam plastics. A specific use of CH₂Cl₂, which seems likely to have increased in recent years, is in the manufacture of hydrofluorocarbons (HFCs) – the non-ozone depleting chemicals used as replacements for CFCs and HCFCs. Given these sources, it is probable that demand for CH₂Cl₂ from developing countries now, and in coming years, will be relatively high. This is supported by elevated levels of CH₂Cl₂ detected over Asia, where Indian emissions are estimated to have increased by two- to fourfold between 1998 and 2008”.

Minor comments, including a few silly word-smithing suggestions:

1. Line 14: “As a consequence” would read better to most

OK, we have changed.

2. Line 32: citations are overall excellent IMHO but for the citation to “ozone recovery is underway”, should consider a citation to Yang et al., JGR, 2008;

<http://onlinelibrary.wiley.com/doi/10.1029/2007JD009675/full>

in addition to the other papers that are cited

OK, we have added in the citation to Yang et al. (2008) where the reviewer requests.

3. Line 49: strike first use of “growth” ... don’t need to use this word twice

OK, we have removed the first “growth”.

4. Line 56: suggest “is” rather than “was”

OK, we have changed this.

5. Line 68: can strike “measurement”

OK, we have removed “measurement”.

6. Line 71: To readers from outside the field, which Nature Communications aspires to reach, it will seem puzzling that the emissions of CH₂Cl₂ can be so much larger than the emissions of CFCs and other ozone-depleting gases, without CH₂Cl₂ being the dominant ODS. Of course, it is the short lifetime of CH₂Cl₂ that account for this so-called puzzle.

I suggest adding a new paragraph between lines 71 and 72, or else a well crated sentence at the end of “gases peaked”, to explain this issue: i.e. try to anticipate some Chemical Engineers in the manufacturing industry struggling with this concept.

OK, we agree with the Reviewer that this could be explained more explicitly. We have added the following sentence to the revised manuscript (see lines 71-74):

“For CH₂Cl₂ and VSLS more generally, relatively large emissions do not have the same impact on atmospheric concentrations, compared to say CFCs, as they are more rapidly oxidized in the troposphere and have much shorter atmospheric lifetimes”.

7. Lines 76 and 79: it cannot be possible that the growth rate of Scenario 1 is 2.85 ppt/year and 4.49 %/year. This can only be true for one year. Same for 6.1 ppt/year and 8.56 %/year for Scenario 2.

Also, one of these numbers make sense given Table S1, where for Scenario 1 between 2080 and 2079, we see a change of 140.1 - 138.4 = 1.7 ppt, and for Scenario 2 we see a change of 300.0 - 296.0 = 4 ppt/year. What pray tell is the assumption for the numerical rise of each scenario? Please get this straight in the paper, and present either as ppt/year or %/year, but not as both.

The numbers of 2.85 ppt/year and 6.1 ppt/year are the growth rates for CH₂Cl₂ at the surface (i.e. the gradients of the straight lines shown in Figure 1c). The quoted %/year numbers were intended to be an approximation based on these values. However, we agree that it is not possible to have both the fixed ppt and the % increases. Therefore, we have deleted the latter as requested.

The numbers given in Table S1 are the amount of CH₂Cl₂ entering the stratosphere under each scenario and this is why they are different.

Finally, Table S2 should extend to an end year consistent with the figures. The table as submitted stops in 2080, but the figures extend past 2080.

OK, we have extended Table S2 to include the full range of data to 2100.

8. Lines 93 and 94: I suggest “By 2050 under Scenario 2, ...”

OK, we have replaced the comma in that sentence.

9. Line 100: suggest a new paragraph at “Under Scenario 2”, or else somewhere in this 21 line paragraph, because the paper will read better with shorter paragraphs.

OK, we have now split this paragraph to improve readability.

10. Line 103: May want to consider adding a citation and a phrase to a paper recently published by Oman et al. in GRL:

<http://onlinelibrary.wiley.com/doi/10.1002/2016GL070471/full>

as well as the Yang et al. paper in ACP on which one of the co-authors participated:

<http://www.atmos-chem-phys.net/14/10431/2014>

OK, this part of the text focuses on changes affecting future ozone. Therefore, we feel that Yang et al. is an appropriate citation to include here (Oman et al. has also been added, but later – please see below).

11. Line 119: I suggest a new paragraph at “Figure 4 shows” and a sentence describing what is depicted in the figure, before the figure is interpreted

OK, we have split this paragraph to improve readability and added a brief sentence describing the figure before the interpretation.

12. Line 134: can strike “already” ... word is not needed and detracts from the message

OK, we have removed “already”.

13. Line 142: might consider citing Oman et al., GRL, 2016 along with ref #27

OK, we have added the citation to Oman et al.(2016).

14. Lines 149 and 152: paper will read better using “growth of CH₂Cl₂”

OK, we have changed this wording.

15. What are the red and blue circles with error bars, on Figure 1d? These do not seem to be explained.

These are mean values in decadal intervals ±1 standard deviation. We have now included this in the caption.

16. The large headers on the figures will detract from the conveyance of the info in the figures, upon publication. See if these labels can be placed inside the panels of the figures, and try to reduce white space between panels.

We thank the reviewer for these suggestions. We have tried to adhere to the journals formatting requirements. If the paper is accepted, we will seek advice from the editor.

End of review ... thanks for submitting such an easy to review paper, on a Saturday night.

Response to Reviewer #2

We thank the reviewer for his/her comments. These are reproduced below in **bold**, followed by our responses in *red italics*. In the revised manuscript file, all changes are highlighted and new text is filled **yellow**.

Reviewer #2 (Remarks to the Author):

In this manuscript Hossaini and colleagues consider the recently-published increases in atmospheric dichloromethane, CH₂Cl₂, mixing ratios and create future scenarios that outline the potentially substantial impacts of continued increases in CH₂Cl₂ emissions on stratospheric ozone levels and Antarctic ozone hole recovery.

The paper is compact and well written, with well-presented figures. I am happy with the quality of the original data. Figures are easy to understand and the supplementary information is appropriate. However, I feel this paper fails to provide a considerable advance to our understanding of this field.

We are pleased that the reviewer finds the paper well-written, well-presented, with easy to understand figures, and is happy with the quality of the original data. However, we disagree that the paper does not provide a considerable advance, on the basis that our study is the first to examine (i) the impact of CH₂Cl₂ on ozone in the recent past, and (ii) the impact of possible future CH₂Cl₂ growth on future stratospheric chlorine and ozone (considering a range of possibilities for future CH₂Cl₂). More detailed responses to specific points are given below.

The first result presented by the authors is the recent growth in atmospheric CH₂Cl₂ (lines 62-66). This increase has already been characterised and published^{1,2,3}, including in the author's prior work. Their estimated global emission of 1 Tg CH₂Cl₂/year (line 69) is also similar to the NOAA estimate reported in the last WMO report⁴ of 841 ±183 Gg CH₂Cl₂/year. The similarity is possibly linked to recent period of slow CH₂Cl₂ growth discussed below.

Indeed, we realise that our emission estimate of 1 Tg CH₂Cl₂/year for the present day is in agreement with other recent independent estimates. This is shown in Figure 2 and we make the point in the text already (see Methods).

The authors then go on to present two future emission scenarios, both based on "...the extrapolation of observed long-term surface trends" (lines 73-74). Scenario 1 is the more conservative, assuming CH₂Cl₂ continues to increase at the mean rate it has done 2004-2014. Scenario 2 is more extreme, considering a projected increased based on the fast growth observed between 2012 and 2014. I would contest that, considering: (1) CH₂Cl₂ emissions vary considerably year on year (Fig. 1b and line 65) and (2) that in the past three years or so the CH₂Cl₂ growth rate has slowed (Fig. 1a), both scenarios could be over estimations and the authors should address this uncertainty and potentially revise these scenarios accordingly.

Note that we actually consider three future CH₂Cl₂ scenarios – including a 'no growth' one as a reference. Regarding these scenarios, in the absence of reliable bottom-up emission estimates, with sufficient coverage for the global industrialised regions, future projections can only be based on observed atmospheric trends, at present. We emphasise that we do not assert that that a particular growth path will be followed. Rather, our scenarios are designed to explore a range of possibilities in terms of future atmospheric CH₂Cl₂ concentrations, and to assess the impact on ozone. We have amended the text in the revised manuscript to make this point clear (for example,

see revised paragraph beginning on line 90 of the marked up revised manuscript, and the penultimate paragraph of the Discussion section).

While CH₂Cl₂ growth has slowed in the last few years, the long-term trend is very clearly upwards. Furthermore, we note that the NOAA surface data does not sample in Asia, where elevated levels of CH₂Cl₂ (and indeed other Cl-VSLS) have been detected (Leedham Elvidge et al. 2015). This paper reported that Indian emissions increased by two- to four-fold over the 1998 to 2008 period. Given the importance of this topic (CH₂Cl₂ not controlled by the Montreal Protocol) and the fact that demand for CH₂Cl₂ could very likely be coming from developing countries (now and in the future), investigation into the impact of possible future CH₂Cl₂ growth on ozone is clearly warranted. The first paragraph of the Results section has been revised to include the above points.

I draw particular attention to point (2) and the slowing of the growth rate in Fig. 1a since 2013. Data in this figure come from NOAA measurements and a recent (31st Oct 16) check of the NOAA data freely available on <http://www.esrl.noaa.gov/gmd/hats/gases/CH2Cl2.html> suggests that this plateau in atmospheric surface CH₂Cl₂ concentrations is continuing. The average CH₂Cl₂ concentration for the first half of 2016 for the five stations between 30-60 °N (to fit with Fig 1a) is ~64 ppt⁵, similar to results from 2013-2015. To ignore this three to four year pattern when creating future projections is potentially misleading.

This is part of an important fact about CH₂Cl₂ and other VSLS Cl species: these species have short (~5 months for CH₂Cl₂⁴) lifetimes and so changes in emissions will have relatively rapid effects on atmospheric abundance. Unlike CFCs, which the authors make comparisons to (e.g. line 70), the tropospheric chlorine burden created by a few years of strong emissions and growth can, relatively quickly, be reversed. This has two impacts for this work:

- **As discussed above, both Scenario 1 and Scenario 2 are inferred from current situations that may already be changing.**
- **Unlike longer-lived gases, where changes in sinks could impact recovery back to pre-industrial atmospheric concentrations, the driving force in future atmospheric CH₂Cl₂ concentrations will be industrial emissions, a factor the authors admit are "poorly characterized" (line 52). I feel that the two limited scenarios discussed by the authors are inadequate for covering the potential fluctuations in anthropogenic supply and demand, especially considering the variations in growth rate shown over the past decade (Fig. 1b).**

It is correct that growth has slowed in the last two years. However, note we cannot consider an incomplete 2016 data record (first half of the year only) given the strong seasonal cycle of CH₂Cl₂ (missing winter months, when lifetime is longer in the NH – see Ch. 1 of WMO 2014). Based on data over the last decade or so, the longer-term trend is clearly upward. As previously noted our scenarios are designed to examine the impact of a range of possible future CH₂Cl₂ concentrations on ozone. Importantly, as noted above, we already consider a no future growth situation in our manuscript, which the reviewer did not acknowledge in his/her review (see Discussion section). We find a delay in ozone return can be expected even if CH₂Cl₂ loading is fixed at present day levels (i.e. if the 'plateau' continues).

In the revised manuscript we have made the no future growth scenario more prominent (see revised text in third paragraph of Results and penultimate paragraph of Discussion). Three scenarios are therefore discussed: the long-term mean growth projected into the future, a higher growth scenario, and a lower growth scenario (no further increases). The text has been amended in numerous places to reflect this change of emphasis. We note that in the absence of any new

policy controls on CH_2Cl_2 , the actual future trend of this gas will be somewhere between the different scenarios discussed here (a point we now make).

In addition to considering a range of future possibilities for CH_2Cl_2 (no future growth, moderate growth, extreme growth), we find a linear response between ozone and CH_2Cl_2 growth (Fig S3). This means that we can easily interpolate to any other scenario that falls within the above range and thereby make predictions. This linear response is important to demonstrate using a detailed atmospheric chemistry model and provides a framework for understanding the impact on ozone of any other assumed CH_2Cl_2 scenario.

Importantly, we also emphasise that our paper shows that CH_2Cl_2 has already had a significant impact on ozone that has increased over the last decade. This result is clearly shown in Figure S2 and Figure 4, is discussed in the text, and is included in the abstract. This result is also clearly independent of any assumed future growth scenario for CH_2Cl_2 , and is important as CH_2Cl_2 is an ozone-depleting gas not controlled by the UN Montreal Protocol.

The authors next go on to discuss model results showing that increasing CH_2Cl_2 emissions into the stratosphere based on increasing surface concentrations are relatively simple. A 'back of the envelope' calculation where I took the average NOAA 2016 CH_2Cl_2 mixing ratio of 64 ppt and projected this forward to 2050 with an increase of 2.85 ppt/year (manuscript Scenario 1, lines 75-76) and then reduced the 2050 value by 18% (representing the 18% decrease in $[\text{CH}_2\text{Cl}_2]$ observed between the marine boundary layer and the layer of zero radiative heating given in both the 2010 and 2014 WMO reports^{4,6}) gave me a rough value of ~133 ppt CH_2Cl_2 entering the stratosphere in 2050. This is around their lower estimation (Fig. 1d).

We are pleased that the reviewer's 'back of the envelope' calculation seems consistent with our model (note previous comment about seasonal cycle of CH_2Cl_2 and 2016 data).

To use this paper to move this research field on from the previous observational work discussed previously^{1,2,3} it would be nice to see the scenarios used to constrain limiting factors we are not able to easily calculate without a model.

The reviewer does not acknowledge that we have quantified the impact of CH_2Cl_2 on ozone in the recent past, and in the future (given a range of possibilities for future CH_2Cl_2). This is an important point to miss out as it is the novel, policy-relevant, and main finding of this study that is not possible to do without a global atmospheric chemistry model.

We stress that the main point of the paper is not the projection of CH_2Cl_2 , but to understand how increases in CH_2Cl_2 have impacted ozone to date, and how they could impact ozone in the future (both magnitude and location). These influences are not readily calculable from a back-of-the-envelope approach.

For example, the authors mention the potential impact of other physical and chemical changes on stratospheric $\text{Cl}_y^{\text{VSLs}}$ and ozone depletion, such as climate-driven changes to stratospheric temperature and circulation (lines 127-128) and tropospheric-stratospheric exchange. The impact of changes in these areas need to be considered when making future predictions of the kind discussed here.

Our approach was designed to consider the impact of past and potential future CH_2Cl_2 growth on ozone and ozone recovery in the Antarctica Ozone Hole region separate from other influences. In this region, ozone trends are dominated by changes in the stratospheric halogen loading (e.g.

(Eyring et al., 2007, JGR; Eyring et al., 2010, ACP; Chapter 9 of SPARC CCMVal, 2010). This seems a reasonable step forward particularly since it has not been done previously, the influences are potentially significant, and CH₂Cl₂ is not controlled by any international protocol. Mention is made of the other factors that will influence ozone recovery to provide context to the new results. It is beyond the scope of this work to consider all potential interacting factors and scenarios (for GHGs etc.,) to attempt to predict actual ozone hole recovery dates. Our aim is to diagnose the impact of CH₂Cl₂ growth using a CTM with realistic meteorology (e.g. polar temperatures). Coupled chemistry-climate model assessments are much more complicated and indirect (e.g. requiring multi-member ensembles and statistical analysis to isolate different factors) and the models will not necessarily simulate realistic stratospheric conditions.

In summary, the authors present previously published data combined with somewhat-limited model projections. Whilst the current and future significance of non-Montreal Protocol regulated VSLs Cl species, such as CH₂Cl₂, is an important topic for debate I feel that this paper does not do enough to address the key limitations in our current understanding of this topic.

This paper includes an update of the NOAA measurement record (two years of data), but more importantly is focussed on a new result, and that is the calculation of ozone sensitivity to CH₂Cl₂ increases. We view this as a novel and important result (as did reviewer #3 explicitly in his/her assessment – and it also appears implicit in reviewer #1's overall assessment of our paper).

Response to Reviewer #3

We thank the reviewer for his/her comments. These are reproduced below in **bold**, followed by our responses in *red italics*. In the revised manuscript file, all changes are highlighted and new text is filled **yellow**.

Reviewer #3 (Remarks to the Author):

Review of The increasing threat to stratospheric ozone from dichloromethane by Hossaini et al.

This manuscript represent a thorough and vital analysis of an emerging atmospheric trace gas species of concern that has large implications for stratospheric ozone recovery. To date this analysis has been missing from the literature.

We thank for the reviewer for these remarks and are delighted with the supportive comments.

Very short-lived substances are not regulated under the current Montreal protocol framework for the protection of stratospheric ozone. Hence, the findings that a delay in ozone recovery of 30 years if the growth rate continues as present is of relevance to a wide community. In a more aggressive growth scenario, ozone recovery in the 21st century is precluded. With approximately 50% of the climate change experienced in spring and summer in the Southern hemisphere over the past 30 years attributed to the springtime Antarctic Ozone hole, a delay in stratospheric ozone recovery has major implications for the climate system. Therefore, this research has broad readership from policy, climate, atmospheric chemistry, and epidemiology interests and I recommend its publication in Nature communications. I have some questions and suggestions that I list below.

We are pleased that the Reviewer notes the broad significance of this work and its relevance across a range of fields.

The title could have more impact representing the implications of this work - delay in ozone recovery could be included.

We thank the Reviewer for this suggestion. If the paper is accepted, we will seek advice from the editor.

The writing in places is a little awkward - 'In consequence' is not an accessible construct - I suggest changing both instances (lines 14 and 35).

OK. We have changed this in both instances.

Line 42: qualify with stratosphere 'minor role in stratospheric ozone...'

OK. We have added "stratospheric".

More examples of the diverse uses of dichloromethane, and perspective on alternatives would add to the introduction (paragraph beginning line 37) - i.e. why has the growth occurred in this species?

OK. In response to this comment and a comment from Reviewer #1, we have added additional text to the first paragraph of the Results section (on industrial sources of CH₂Cl₂). This text points out that CH₂Cl₂ is used in the production of HFCs, a source which may have increased in recent years.

Line 44 suggest replacing 'due to expected relatively ' with 'therefore'

OK. We have made this change.

Line 67 it could be interesting to note here (like most CFCs), while the emissions are weighted toward the Northern hemisphere, the consequences are disproportionately experienced in the Southern hemisphere.

Thank you for this suggestion. As the text in this paragraph has now changed in response to another reviewer's comment, we feel that this point may feel out of place in this section.

Line 79 - sentence The 2012-2014 could be removed - this point has been made in the previous sentence.

OK. We have removed the second "2012-2014" on that line.

The point that your model accurately simulates both the surface and the UTLS CH₂Cl₂ observations providing confidence that the emissions and lifetimes are well represented needs to be made more clearly (paragraph beginning line 72).

OK. We have amended the last sentence of that paragraph to:

"Critically, the model reproduces well observed levels of CH₂Cl₂ around the tropopause in the recent past (Fig. 1d inset) and, therefore, the stratospheric chlorine perturbation in response to increasing surface CH₂Cl₂ concentrations is realistic in our simulations".

Line 89 write the stratospheric lifetime here (and reference)

Ok, we have added the approximation stratospheric lifetime in here as calculated by our model (of the order of 1-2 years).

Paragraph beginning line 87 - what is the role of chlorine introduced to the lower stratosphere under high volcanic aerosol loading? If you were able to quantify this with an additional scenario run it would add to the future possible implications of this work.

It is expected that halogen-driven ozone loss is generally enhanced for a period post volcanic eruptions. Quantifying the possible impact of a future volcanic eruption we feel is beyond the scope of this study and would detract from the main messages about increasing CH₂Cl₂.

Line 132. The climate effect of the Antarctic ozone hole is outlined in chapter 4 of the 2014 WMO report - the text around this could be tightened significantly.

OK, we have included a citation to the WMO chapter and also now note that ozone depletion may potentially impact trends in sea ice extent. Obviously there is a lot to potentially say here but we are trying to keep the text concise as not to dilute the CH₂Cl₂-focused message.

Line 161 Note that under growth scenario 2 no recovery is seen by 2100

OK, we now make that point in the second paragraph of the Discussion section.

Line 167 the strengthening of the Brewer Dobson circulation is also not included.

Ok, we have added "stratospheric dynamics" to this list.

Line 213 include the reference for 0.43 year lifetime.

This is calculated from our modelling work. We have now noted this.

Line 217 what is the uncertainty introduced in your methodology by using modern day meteorology? It should be possible to quantify this by having simulation with GCM evolving meteorology including a stronger BD circulation, and cooler stratospheric conditions.

We deliberately chose to use present day meteorology within a CTM framework because that way future ozone changes in our simulations are solely due to changing stratospheric composition, with and without CH₂Cl₂. This approach is advantageous as it allows us to isolate a clean signal of the impact of CH₂Cl₂ growth on ozone which, we feel, is an important policy-relevant result. Mention is made of the other factors that will influence ozone recovery to provide context to the new results. Assessment of these would require the much more complicated assessment of chemistry-climate model simulations (e.g. statistical analysis of different multi-member ensemble simulations). See also the response to Reviewer 2.

Reviewers' comments:

Reviewer #1 (Remarks to the Author):

This paper is excellent in my opinion and acceptable for publication in Nature Communications in its present form. Thanks for making so many changes in reply to my earlier comments. I have also read the other reviews as well as the responses to these comments: again, kudos to the team for making so many changes (albeit, mainly minor) to these comments. With regards to the notion expressed by Reviewer #2 that the paper is not a particularly strong advance of our present knowledge, I feel that quantification of the effect of a very short lived anthropogenic chlorocarbon on stratospheric ozone is new and important, and I believe the team has done a nice job of tying their model calculations to atmospheric observations in a manner that is entirely appropriate for this journal.

Two more comments, both extremely minor:

1) For the new sentence:

For CH₂Cl₂ and VSLs more generally, relatively large emissions do not have the same impact on atmospheric concentrations, compared to say CFCs, as they are more rapidly oxidized in the troposphere and have much shorter atmospheric lifetimes".

please consider writing "say CFCs, as CH₂Cl₂ is more rapidly ..." for clarity.

2) If the comments from Robyn Schofield and the other two anonymous reviewers helped improve the manuscript, might want to add a phrase in the acknowledgements :).

For some reason, even though I have submitted two consecutive very positive reviews, I choose to remain anonymous. Trust me, two consecutive positive reviews is not my norm, so you'd likely be surprised by my identity. Anyway, I have grown to appreciate the value of the anonymous peer-review process, so will honor the "anonymous" part of this phrase.

Reviewer #2 (Remarks to the Author):

none

Reviewer #3 (Remarks to the Author):

The authors have made many additions to their manuscript in response to the comments made by the reviewers. In a couple of places however, the changes have resulted in reduced clarity or errors.

There seems to be actually three scenarios run, a no future growth, future growth at 2.85 ppt/year and rapid growth at 6.1 ppt/year. For clarity either these should be given as scenarios 1, 2 and 3 and the no future growth scenario is added to Figures 1 and 3. Alternatively the no future growth scenario introduced is removed from the paper as it is confusing with the no CH₂Cl₂ reference run (and seems only to be discussed in reference to currently delaying the ozone hole recovery by four years currently). The sentence introduced in line 101 of the highlighted text confusingly states that the no further CH₂Cl₂ growth is the reference. I think it is an interesting scenario, and believe that it should be more carefully integrated into the paper than it is currently done.

Secondly, the sentence added concerning sea-ice from Arblaster is incorrect: from the executive summary of Arblaster:

"The influence of stratospheric ozone depletion on Antarctic sea ice increases reported in the last Ozone Assessment is not supported by a number of new coupled modeling studies. These suggest

that ozone depletion drives a decrease in Southern Hemisphere sea ice extent and thus did not lead to the small observed increase. However, there is low confidence in this model result because of large uncertainties in the simulation of Antarctic sea ice."

How the authors responded to my original review comment to tighten the text by adding the WMO reference is disappointing. Please remove the sentence, and simply replace references 32-34 with reference 35 as it is the most up to date review of the climate response to ozone depletion.

I do not believe that an evolving meteorology run, which would represent a more realistic future meteorology, say CCMVal multi-model mean from REF2 or taking single model meteorology from an IPCC run i.e. a RCP6.0 run, is beyond the scope of this paper, or need to be time consuming. I am confident that this modelling team has a realistic future climatology that would be appropriate to use at their disposal. As both reviewer 2 and myself raised this issue I believe it is warranted.

Response to Reviewer #1

We thank the reviewer for his/her comments. These are reproduced below in **bold**, followed by our responses in *red italics*. In the revised manuscript file, all changes are highlighted and new text is filled **yellow**.

Reviewer #1 (Remarks to the Author):

This paper is excellent in my opinion and acceptable for publication in Nature Communications in its present form. Thanks for making so many changes in reply to my earlier comments. I have also read the other reviews as well as the responses to these comments: again, kudos to the team for making so many changes (albeit, mainly minor) to these comments. With regards to the notion expressed by Reviewer #2 that the paper is not a particularly strong advance of our present knowledge, I feel that quantification of the effect of a very short lived anthropogenic chlorocarbon on stratospheric ozone is new and important, and I believe the team has done a nice job of tying their model calculations to atmospheric observations in a manner that is entirely appropriate for this journal.

Thank you for the very positive and supportive remarks regarding our paper. We agree that our findings on the threat to ozone posed by CH₂Cl₂ growth are important.

Two more comments, both extremely minor:

1) For the new sentence:

"For CH₂Cl₂ and VSLS more generally, relatively large emissions do not have the same impact on atmospheric concentrations, compared to say CFCs, as they are more rapidly oxidized in the troposphere and have much shorter atmospheric lifetimes"

please consider writing "say CFCs, as CH₂Cl₂ is more rapidly ..." for clarity.

OK, we have amended the text as requested.

2) If the comments from Robyn Schofield and the other two anonymous reviewers helped improve the manuscript, might want to add a phrase in the acknowledgements.

For some reason, even though I have submitted two consecutive very positive reviews, I choose to remain anonymous. Trust me, two consecutive positive reviews is not my norm, so you'd likely be surprised by my identity. Anyway, I have grown to appreciate the value of the anonymous peer-review process, so will honour the "anonymous" part of this phrase.

Indeed, we believe that the paper has been strengthened as a result of all of the reviewer comments and we will certainly acknowledge this if the paper is accepted.

Response to Reviewer #3

We thank the Dr Schofield for her comments. These are reproduced below in **bold**, followed by our responses in *red italics*. In the revised manuscript file, all changes are highlighted and new text is filled **yellow**.

Reviewer #3 (Remarks to the Author):

The authors have made many additions to their manuscript in response to the comments made by the reviewers. In a couple of places however, the changes have resulted in reduced clarity or errors.

We are very grateful to Dr Schofield for providing comments that have ultimately strengthened the paper and for allowing us to clarify how the sensitivity of our results to different future meteorology could be addressed in the most scientifically robust manner possible.

There seems to be actually three scenarios run, a no future growth, future growth at 2.85 ppt/year and rapid growth at 6.1 ppt/year. For clarity either these should be given as scenarios 1, 2 and 3 and the no future growth scenario is added to Figures 1 and 3. Alternatively the no future growth scenario introduced is removed from the paper as it is confusing with the no CH₂Cl₂ reference run (and seems only to be discussed in reference to currently delaying the ozone hole recovery by four years currently). The sentence introduced in line 101 of the highlighted text confusingly states that the no further CH₂Cl₂ growth is the reference. I think it is an interesting scenario, and believe that it should be more carefully integrated into the paper than it is currently done.

OK, we agree that the “no future growth” scenario should be labelled in a consistent manner with the others. We also agree that this scenario is interesting and should be given more prominence in the text and figures. As requested, we have made the following changes:

- The no future growth scenario was explicitly run for the revised manuscript (rather than estimated). We now refer to it as “Scenario 3” and have added appropriate text in both the main part of the paper (see lines 102-105 and lines 196-198 of the marked up revised manuscript) and also in the Methods (lines 298-301).*
- We clarified the text so that it is not confused with the reference no CH₂Cl₂ run.*
- We have added the appropriate Scenario 3 lines (dashed orange) to Figures 1, 3, 6 and S1.*

Secondly, the sentence added concerning sea-ice from Arblaster is incorrect: from the executive summary of Arblaster:

“The influence of stratospheric ozone depletion on Antarctic sea ice increases reported in the last Ozone Assessment is not supported by a number of new coupled modeling studies. These suggest that ozone depletion drives a decrease in Southern Hemisphere sea ice extent and thus did not lead to the small observed increase. However, there is low confidence in this model result because of large uncertainties in the simulation of Antarctic sea ice.”

How the authors responded to my original review comment to tighten the text by adding the WMO reference is disappointing. Please remove the sentence, and simply replace references 32-34 with reference 35 as it is the most up to date review of the climate response to ozone depletion.

OK, thank you for pointing this out. As requested we have deleted the line that we previously introduced and replaced those references with Arblaster et al.

I do not believe that an evolving meteorology run, which would represent a more realistic future meteorology, say CCMVal multi-model mean from REF2 or taking single model meteorology from an IPCC run i.e. a RCP6.0 run, is beyond the scope of this paper, or need to be time consuming. I am confident that this modelling team has a realistic future climatology that would be appropriate to use at their disposal. As both reviewer 2 and myself raised this issue I believe it is warranted.

Once again we are grateful to Dr Schofield for allowing us to clarify with her how best to examine the sensitivity of our results to different potential future meteorology. We believe that we have addressed the issue thoroughly and that the manuscript has been strengthened accordingly. We summarise the steps we have taken below.

New SLIMCAT sensitivity runs:

- Originally the manuscript presented SLIMCAT simulations that in the future period used assumed meteorology of the year 2012 (but time-varying in the past). In the revised manuscript, we now examine the sensitivity of future ozone loss due to CH₂Cl₂ to different meteorology, by running the model forward with two additional and different assumed meteorological years. We chose the years 2002 and 2006 as these conditions were somewhat 'remarkable'. For example, 2002 was anomalously warm as a result of a major stratospheric warming and Antarctic column ozone was high compared to previous years (e.g. Sinnhuber et al., 2003). In contrast, the ozone hole in 2006 was relatively large and the winter was cooler.*
- We have added a new figure (Fig 5) to the main manuscript, summarising the results of the above model runs, and appropriate new text (please see line 164 onwards and line 220 onwards in the marked-up version). Absolute ozone decreases due to CH₂Cl₂ are somewhat sensitive to the meteorological conditions, e.g. less loss under the 2002 ("warm") conditions, but the impact of these changes on the ozone return dates was not significant.*
- In the lines given above, we acknowledge that these SLIMCAT runs, although spanning a range of possibilities, do not have future inter-annual meteorological variability and will not capture prolonged climate-induced changes in the future, including changes to the Brewer-Dobson circulation and cooling in the upper stratosphere. However, we have performed new simulations that consider such effects (see below).*

New UMSLIMCAT run with evolving meteorology:

- We have performed a new set of model simulations with/without CH₂Cl₂ using the UMSLIMCAT chemistry-climate model forced by the IPCC RCP 6.0 scenario, as requested. These runs consider evolving meteorology/dynamics in the future. Transient runs covering the period 1970 to 2100 were performed. Text describing the model and these new runs has been added to the Methods section. We used CH₂Cl₂ growth Scenario 1 in the with-CH₂Cl₂ case.*
- A 2nd additional figure (Fig 7) has been added to the main manuscript summarising the UMSLIMCAT results. We find statistically significant (at the 95% level according to a student's t test) column ozone differences between the no CH₂Cl₂ run and the run with CH₂Cl₂ growth, with the largest impact of CH₂Cl₂ over Antarctica, where this impact increases in time – corroborating our CTM findings.*
- From UMSLIMCAT, we find that CH₂Cl₂ growth delays the return of Antarctic column ozone by 17 years. This is smaller than the delay predicted by the CTM (which generally shows much deeper polar ozone loss for a given amount of chlorine), but clearly a delay of more than a decade is very significant. Related to this, we have added text (see lines 232-238) discussing*

what we believe to be an important point: that different models, even when running with the same CH₂Cl₂ growth scenario and for a common 1980 baseline, will predict different delays on the ozone return date due to CH₂Cl₂. For example, those models characterised by an inherent relatively slow rate of ozone return will predict a larger delay due to CH₂Cl₂ because of the diverging stratospheric Cl_y field in time; i.e. between Cl_y in a no CH₂Cl₂ run and one in which sustained CH₂Cl₂ growth occurs.

*Overall, given that the size of the ozone return date delay due to CH₂Cl₂ is more than a decade (predicted by both our models), our results highlight the need for CH₂Cl₂ growth to be considered in future multi-model ozone assessments. **We believe that the paper has been strengthened as a result of the reviewer comments and the above revisions. Our main finding, that CH₂Cl₂ growth is an increasing threat to stratospheric ozone, is now even more robust.***

REVIEWERS' COMMENTS:

Reviewer #3 (Remarks to the Author):

I commend the authors on the work they have done to address my questions around changes in future meteorology. As they note this has only strengthened their conclusions that CH₂Cl₂ should be included in climate simulations and that lack of policy actions will result in a significant delay in the the ozone hole recovery. I think that this paper will be extremely impactful in shaping policy and scientific endeavours in ozone and climate research. I believe the paper to be very polished and ready for publication as is. The new figures and addition of scenario 3 in them has added nice clarity.